# SGMD: Score Gradient Matching Distillation
# for Few-Step Video Diffusion Distillation

**Zhuguanyu Wu** [1 2]   **Ruihao Gong** [1]   **Yang Yong** [2]   **Yushi Huang** [2 3]   **Xiangyu Fan** [2]   **Lei Yang** [2]   **Dahua Lin** [2]
**Xianglong Liu** [1]

## Abstract

Distribution Matching Distillation (DMD) is a widely used paradigm for accelerating inference in few-step video diffusion models. However, DMD-style video distillation faces two coupled challenges: the fake score must track a continuously evolving generator, making training costly when frequent updates are required, while reverse-KL-style matching can be mode-seeking and conservative for preserving strong motion dynamics. To address these issues, we propose **Score Gradient Matching Distillation (SGMD)**. SGMD adopts a fake-score perspective by directly optimizing the fake score toward the teacher, while using teacher stop-gradient Fisher as a stable distribution-matching objective. We provide a gradient analysis that motivates this objective choice under ideal tracking. Building on this, SGMD introduces a pair of dual potentials: negative-residual (NR) for outer-loop correction and residual-contraction (RC) for inner-loop tracking. Empirically, compared to DMD2, SGMD achieves an approximately $\sim 3\times$ training speedup and substantially improves motion dynamics for 4-step distilled models while preserving temporal consistency. A human study confirms that SGMD is preferred in motion quality and overall preference, while visual quality and text alignment remain comparable. Code is available at https://github.com/ModelTC/LightX2V/tree/main/lightx2v_train.

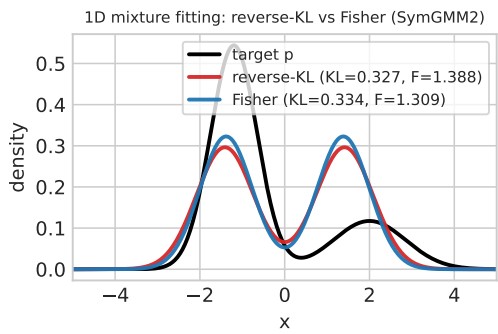

*Figure 1.* **Motivating 1D mixture-fitting example.** Reverse-KL-style matching tends to produce a conservative fit that avoids low-density regions of the target distribution, while Fisher divergence yields a smoother score-matching signal.

## 1. Introduction

Diffusion models (Ho et al., 2020; Song et al., 2020; Geng et al., 2025) have recently achieved remarkable progress in video generation (Wang et al., 2025; Team, 2025; Cai et al., 2025). However, modern video diffusion models are costly to run: large parameter counts, high latent dimensionality, and multi-step sampling all hinder deployment and interactive applications (Ben Yahia et al., 2024). Existing acceleration techniques include quantization and low-bit inference (Li et al., 2025; Huang et al., 2024a; Wu et al., 2025a;b; Huang et al., 2026; Gong et al., 2025a; Li et al., 2021; Gong et al., 2025b; 2019), feature caching (Ma et al., 2024; Liu et al., 2025; Lv et al., 2025; Huang et al., 2025b; Deng et al., 2018; Li et al., 2018; Yang et al., 2019; 2021), parallel sampling (Shih et al., 2023; Fang et al., 2024; FlagOS Team, 2025), and few-step distillation (Yin et al., 2024b;a; Zhou et al., 2024; Frans et al., 2025). Among them, few-step distillation is particularly attractive for video generation because it directly reduces sampling steps while largely preserving the original architecture and deployment pipeline.

Distribution Matching Distillation (DMD) and its variants (Yin et al., 2024b;a; Fan et al., 2025) are a strong line of work for few-step video diffusion distillation via distribution matching. In aggressive few-step regimes, however, DMD-style training faces two practical challenges. First, it becomes a two-timescale problem: the student-side aux-

---

[1]Beihang University [2]SenseTime Research [3]Hong Kong University of Science and Technology. Correspondence to: Ruihao Gong <gongruihao@buaa.edu.cn>.

*Proceedings of the $43^{rd}$ International Conference on Machine Learning*, Seoul, South Korea. PMLR 306, 2026. Copyright 2026 by the author(s).

iliary score network (the fake score) must track a rapidly evolving generator, and maintaining this tracking often requires multiple fake-score updates that dominate training cost. Second, reverse-KL-style matching is mode-seeking (Zheng et al., 2026) and can behave conservatively, as illustrated by Fig. 1, which may suppress motion in motion-rich video generation under few-step distillation.

These challenges motivate us to revisit the distribution-matching objective and the fake-score update mechanism together. We build on teacher stop-gradient Fisher: under ideal tracking, it induces an effective outer-loop update direction consistent with DMD-style distribution matching, while avoiding teacher input gradients and retaining a stable gradient structure.

Departures from this ideal regime are inevitable: the fake score cannot instantly follow a rapidly evolving generator. SGMD adopts a **fake-score perspective** and turns one-sided tracking into cooperative alignment: the fake score moves toward the teacher, while the generator tracks it to maintain score-consistency. We instantiate this idea with dual potentials: negative-residual (NR) corrects the outer-loop generator update, and residual-contraction (RC) contracts the tracking residual in the fake-score update. This makes the Fisher-style signal usable in practice with fewer fake-score updates, yielding a lightweight two-step bilevel update.

In summary, we propose **Score Gradient Matching Distillation (SGMD)** for few-step video diffusion distillation. Our main contributions are as follows:

- We provide a principled motivation for using teacher stop-gradient Fisher as a stable distribution-matching objective under ideal tracking, and analyze how tracking lag bends the net one-iteration update direction.
- We propose Score Gradient Matching Distillation (SGMD), a fake-score perspective with a pair of dual potentials (NR/RC) that decouple outer-loop correction from inner-loop contraction, enabling stable, low-overhead two-step updates.
- We validate SGMD on large-scale video diffusion distillation with a 14B teacher (Wan2.1-T2V-14B), showing improved motion dynamics and temporal consistency under 4-step distillation, an approximately $\sim 3\times$ training speedup by reducing fake-score updates per iteration, and human-preference gains in motion quality and overall preference while maintaining comparable visual quality and text alignment.

## 2. Preliminaries

### 2.1. DMD

Distribution Matching Distillation (DMD) considers a generator $G_\theta$ inducing a distribution $q$, a fixed diffusion teacher

providing the target score $s_{\text{real}}$ w.r.t. the target distribution $p$, and an auxiliary student-side score network (*i.e.*, fake score) $s_{\text{fake}}$ that approximates the score of $q_\theta$. Throughout, the teacher network $\mu_{\text{base}}$ is kept frozen. The generator training target is given as:

$$\nabla_\theta D_{\text{KL}}(q\|p) = \mathbb{E}_{\substack{z\sim\mathcal{N}(0;\mathbf{I})\\x=G_\theta(z)}}\left[\left(s_{\text{fake}}(x) - s_{\text{real}}(x)\right)\frac{dG}{d\theta}\right], \tag{1}$$

where the scores are given as:

$$s_{\text{fake}}(x_t, t) = \frac{\alpha_t\mu_\psi(x_t, t) - x_t}{\sigma_t^2}, \tag{2}$$

$$s_{\text{real}}(x_t, t) = \frac{\alpha_t\mu_{\text{base}}(x_t, t) - x_t}{\sigma_t^2}. \tag{3}$$

The fake score's training target is given as:

$$\mathcal{L}(\psi) = \|\mu_\psi(x_t, t) - x_0\|^2 \tag{4}$$

where $x_t$ is obtained by the standard forward noising process:

$$x_t = \alpha_t x_0 + \sigma_t\epsilon. \tag{5}$$

Training is typically performed in an alternating, two-timescale manner: the inner loop regresses $\psi$ to closely follow the rapidly changing generator, while the outer loop updates $\theta$ using a distribution-matching objective assuming $\psi$ is sufficiently up-to-date. This structure exposes a practical bottleneck: maintaining small tracking error often requires multiple fake-score updates per iteration (high training cost), whereas reducing inner-loop updates leads to tracking lag, instability, and degraded sample consistency. SGMD addresses this bottleneck by enabling stable following and effective outer-loop updates with substantially lower overhead.

### 2.2. SIM

Score Implicit Matching (SIM) (Luo et al., 2024) takes the Fisher Divergence as the training object:

$$\mathcal{L}(\theta, \psi) = \frac{1}{2}\|s_{\text{fake}}(x_t) - s_{\text{real}}(x_t)\|^2 \tag{6}$$

The gradient is given as:

$$\nabla_\theta\mathcal{L}(\theta) = (s_{\text{fake}}(x_t) - s_{\text{real}}(x_t))(\nabla_\theta s_{\text{fake}} - \nabla_\theta s_{\text{real}}) \tag{7}$$

Since the optimal inner solution $\psi^*(\theta)$ depends on $\theta$, let $y(\theta) := \mu_{\psi^*(\theta)}(x_t, t)$. Differentiating $y(\theta)$ includes not only the explicit term propagated through the forward process in Eq. (5), but also an implicit term induced by the variation of $\psi^*(\theta)$:

$$\frac{d}{d\theta}y(\theta) = \underbrace{\frac{\partial\mu_\psi(x_t, t)}{\partial\theta}}_{\text{explicit term}} + \underbrace{\frac{\partial\mu_\psi(x_t, t)}{\partial\psi}\frac{d\psi^*}{d\theta}}_{\text{implicit term}} \tag{8}$$

By substituting Eq. (8) into Eq. (7), SIM derives that the implicit-gradient contribution can be equivalently obtained by minimizing the following loss function:

$$\mathcal{L}^{(2)}(\theta) = \Big\langle s_{\text{fake, sg}[\theta]}(x_t, t) - s_{\text{real}}(x_t, t),$$
$$s_t(x_t|x_0) - s_{\text{fake, sg}[\theta]}(x_t, t) \Big\rangle. \quad (9)$$

where sg[·] means stop-gradient operation, and $s_t(x_t|x_0) = (\alpha_t x_0 - x_t)/\sigma_t^2$.

For a clearer derivation, we rewrite the implicit loss term $\mathcal{L}^{(2)}(\theta)$ in the $x$-prediction form:

$$\Delta_t = \mu_{\psi,\text{sg}[\theta]}(x_t, t) - \mu_{\text{base}}(x_t, t),$$
$$r_t = x_0 - \mu_{\psi,\text{sg}[\theta]}(x_t, t),$$
$$c(t) = \alpha_t^2/\sigma_t^4 \quad (10)$$
$$\mathcal{L}^{(2)}(\theta) = c(t)\, \Delta_t^\top\, r_t.$$

The explicit term can be written as:

$$\mathcal{L}^{(1)}(\theta) = \frac{1}{2} c(t) \left\| \Delta_t \right\|^2. \quad (11)$$

and the overall loss is given as:

$$\mathcal{L}_{\text{SIM}} = \mathcal{L}^{(1)}(\theta) + \mathcal{L}^{(2)}(\theta) + \mathcal{L}(\psi) \quad (12)$$

## 3. Method

In this section, we first introduce *teacher stop-gradient Fisher divergence* as a stable distribution-matching objective that avoids unreliable teacher input gradients. We then present the *fake-score perspective*, which reframes training as directly improving the fake score toward the teacher while using the generator as a tracker. Next, we provide a *gradient analysis* to show how tracking lag bends the effective one-iteration update direction and motivates explicitly correcting this effect. Finally, we introduce **SGMD**, which instantiates this perspective with dual potentials (NR/RC) and a lightweight two-step update to achieve stable training with low fake-score update overhead.

### 3.1. Teacher Stop-Gradient Fisher Divergence

A direct approach is to apply standard score matching and backpropagate teacher input gradients through $x_t$. However, we empirically observe that training becomes extremely unstable once teacher input gradients are enabled. We attribute this to the fact that, during distillation, $x_t$ is often induced by generated samples; teacher input gradients on out-of-distribution (OOD) states can be unreliable and then amplified by backpropagation. We therefore adopt the teacher stop-gradient Fisher objective as a stable alternative:

$$\mathcal{L}_{\text{Fisher}}(\theta, \psi) := \frac{1}{2} \left\| s_{\text{fake}}(x_t, t) - s_{\text{real}}(\text{sg}[x_t], t) \right\|^2$$
$$= \frac{1}{2} c(t) \left\| \Delta_t \right\|^2. \quad (13)$$

Under ideal conditions, Eq. (13) induces the same descent direction as reverse-KL-style distribution matching in DMD (Proposition 3.1).

**Proposition 3.1.** *Assume ideal tracking: the learned fake score and teacher score equal the ground-truth scores, i.e.,*

$$s_{fake}(x_t, t) = s_{q_{\theta,t}}(x_t), \qquad s_{real}(x_t, t) = s_{p_t}(x_t).$$

*Then the effective outer-loop descent direction induced by one-sided Fisher is directionally consistent with that induced by reverse-KL-style distribution matching (i.e., $\text{KL}(q\|p)$), and both align with the score difference $s_{fake} - s_{real}$.*

Under this teacher stop-gradient setting, we still encounter the following training difficulties:

- Using the explicit loss $\mathcal{L}^{(1)}(\theta)$ alone: the fake score fails to continuously track the rapidly evolving generator, and the distillation quality tends to degrade after some training.
- Using $\mathcal{L}^{(1)}(\theta)$ together with $\mathcal{L}^{(2)}(\theta)$ as in SIM: although tracking can be improved, the generator often fails to effectively converge for large-scale models, leading to blurry and less sharp outputs.

### 3.2. Fake-Score Perspective

Most prior DMD-style methods adopt a generator perspective: the fake score is treated as an auxiliary tracker that is repeatedly fit to the current generator-induced distribution, and the generator is updated assuming the tracker is sufficiently up-to-date. In contrast, SGMD adopts a fake-score perspective: we treat the fake score as the primary optimization target that should move toward the teacher score. Crucially, the fake score is a coupled object that depends on both networks, which we write as $s_{\text{fake}}(\cdot, t; \psi, \theta)$: $\psi$ parameterizes the fake-score model, while $\theta$ determines the generator-induced distribution on which the score is defined. Meanwhile, the generator acts as a tracker that maintains score-consistency, ensuring compatibility between the current generator-induced distribution and the learned fake score. We provide a formal justification of this perspective in Appendix A.1.

This perspective suggests analyzing objectives through the net update direction of one coupled iteration, which we do next in Sec. 3.3.

### 3.3. Gradient Analysis

We analyze objectives through the fake-score perspective and focus on the effective update direction on the generator output $x_0$ induced by one iteration of the coupled optimization. Let $\mu_{\text{fake}}(x_t, t) \equiv \mu_{\psi,\text{sg}[\theta]}(x_t, t)$ be the fake-score $x_0$-prediction and $\mu_{\text{real}}(x_t, t) \equiv \mu_{\text{base}}(x_t, t)$ the paired target prediction; $x_t$ is obtained by Eq. (5). Fig. 2a–c visualize

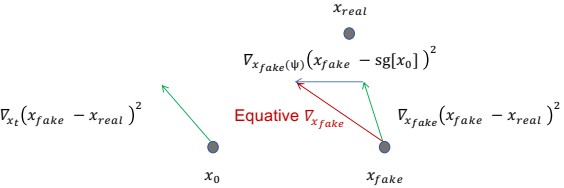

*(a)* Teacher stop-gradient Fisher: the distribution-matching term can be bent in the coupled system (biased net direction on $x_{\text{fake}}$).

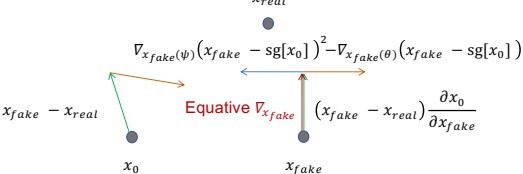

*(b)* SIM: the induced net direction can become conservative due to tracking-induced terms.

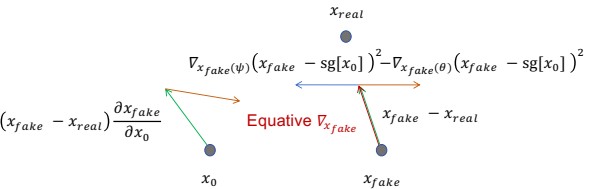

*(c)* SGMD (ours): NR and RC dual potentials decouple correction vs. contraction to recover the desired direction.

*Figure 2.* Gradient behaviors under the fake-score perspective. Arrows show the net one-iteration direction on $x_{\text{fake}}$ induced by coupled $(\theta, \psi)$ updates: Fisher can be bent by coupling-induced tracking lag; SIM may become conservative; SGMD restores the desired direction via NR and RC.

the resulting behaviors for Fisher, SIM, and SGMD. Empirically, high-quality few-step distillation benefits from the alignment condition

$$\nabla_{x_{\text{fake}}} \mathcal{J}(\theta, \psi) \propto \mu_{\text{fake}}(x_t, t) - \mu_{\text{real}}(x_t, t), \qquad (14)$$

where $\mathcal{J}(\theta, \psi)$ denotes the objective induced by one coupled iteration. Importantly, we do not require a closed-form expression of $\mathcal{J}$; throughout this section it serves as a shorthand for the composition of the $\theta$-update and the $\psi$-update under teacher stop-gradient control, and we only analyze its induced net direction on $x_{\text{fake}}$. Systematic deviation from Eq. (14) leads to conservative updates and less vivid motion and details.

### 3.3.1. FISHER SCORE MATCHING

We first revisit Fisher-style score matching under the coupled training dynamics. In distillation, we approximate the two scores by $s_{\text{fake}}(x_t, t)$ and $s_{\text{real}}(x_t, t)$ and update the generator with a teacher-aligned Fisher objective (Eq. (13)) while treating $\psi$ as fixed. This yields a clean outer direction on $x_0$. However, the iteration does not end there: after updating $\theta$, we must update $\psi$ to re-fit the fake score to the new generator-induced distribution. This tracking step changes

the score field and therefore bends the net one-iteration effect on $x_0$, making the overall update deviate from the original Fisher direction, as shown in Fig. 2a.

### 3.3.2. SIM

For brevity, we write $\mu_{\text{fake}}(x_t, t)$ as $x_{\text{fake}}$ and $\mu_{\text{real}}(x_t, t)$ as $x_{\text{real}}$ in the following. The SIM objective combines an explicit Fisher term $\mathcal{L}^{(1)}(\theta)$ and an implicit term $\mathcal{L}^{(2)}(\theta)$. In the $x$-prediction form, their sum can be rewritten as (up to terms independent of $\theta$):

$$\begin{aligned}
\mathcal{L}_{\text{SIM}}(\theta) &= \mathcal{L}^{(1)}(\theta) + \mathcal{L}^{(2)}(\theta) \\
&= c(t)\Big(\tfrac{1}{2}\|x_{\text{real}}\|^2 - \tfrac{1}{2}\|x_{\text{fake}}\|^2 \\
&\qquad + \langle x_0, x_{\text{fake}}\rangle - \langle x_0, x_{\text{real}}\rangle\Big),
\end{aligned} \qquad (15)$$

which yields the following effective direction on $x_{\text{fake}}$:

$$\nabla_{x_{\text{fake}}} \mathcal{L}_{\text{SIM}} = c(t)\Big((x_0 - x_{\text{fake}}) + \Big(\tfrac{dx_0}{dx_{\text{fake}}}\Big)^* (x_{\text{fake}} - x_{\text{real}})\Big). \qquad (16)$$

Here $(\cdot)^*$ denotes the adjoint (VJP) operator under the Frobenius inner product, and $dx_0/dx_{\text{fake}}$ denotes a $\theta$-induced directional map. Eq. (16) decomposes SIM into two behaviors: the first term encourages tracking and is opposite to the regression gradient used to train the fake score; the second term aligns with reverse-KL-style distribution matching (i.e., $\text{KL}(q\|p)$). Thus, SIM may improve tracking. However, the tracking-induced term biases the net direction on $x_{\text{fake}}$, especially when tracking lag persists (Fig. 2b).

These observations suggest two requirements: enforcing the desired outer direction in Eq. (14), and maintaining tracking so the fake score remains compatible with the evolving generator. We next show how SGMD instantiates them with a simple two-step update.

### 3.4. SGMD

SGMD is a simple two-step update scheme that enforces the desired outer direction in Eq. (14) when updating the generator, while maintaining tracking so the fake score stays compatible with the evolving generator. Concretely, we decompose the tracking problem into two complementary roles: an outer-loop direction correction applied to the generator update, and an inner-loop residual contraction applied to the fake score update.

### 3.4.1. DUAL POTENTIALS

Let $x_0$ denote the generator sample and let $x_t$ be its noisy state at noise level $t$ obtained by Eq. (5). We use $\text{sg}[\cdot]$ to denote stop-gradient. Define the tracking residual

$$r(x_0, x_t) := \text{sg}[x_0] - x_{\text{fake}}. \qquad (17)$$

---

**Algorithm 1** SGMD training

---

1: **Input:** generator $G_\theta$, fake score model $\mu_\psi$, real score model $\mu_{\text{base}}$, coefficient $\lambda$
2: **for** each training iteration **do**
3:     Sample a minibatch; draw noise level $t$ and $\epsilon \sim \mathcal{N}(0, I)$
4:     Generate $x_0 \leftarrow G_\theta(\cdot)$ and form $x_t \leftarrow \alpha_t x_0 + \sigma_t \epsilon$ (Eq. (5))
5:     Compute $x_0$-predictions $x_{\text{fake}} \leftarrow \mu_\psi(x_t, t)$ and $x_{\text{real}} \leftarrow \mu_{\text{base}}(\text{sg}[x_t], t)$; set $\Delta_t \leftarrow x_{\text{fake}} - x_{\text{real}}$
6:     Compute Fisher loss: $\mathcal{L}_{\text{Fisher}}(\theta) \leftarrow \frac{1}{2} c(t) \|\Delta_t\|^2$ with $c(t) = \alpha_t^2/\sigma_t^4$ (Eq. (13))
7:     Compute Residual: $r \leftarrow \text{sg}[x_0] - x_{\text{fake}}$ (Eq. (17))
8:     Compute Dual potentials: $\mathcal{L}_{\text{NR}}(\theta) \leftarrow -\frac{1}{2}\|r\|^2$, $\mathcal{L}_{\text{RC}}(\psi) \leftarrow +\frac{1}{2}\|r\|^2$ (Eq. (18))
9:     **Update Generator** with $\psi$ detached: $\theta \leftarrow \theta - \eta_\theta \nabla_\theta(\mathcal{L}_{\text{Fisher}}(\theta) + \lambda \mathcal{L}_{\text{NR}}(\theta))$
10:    **Update Fake-score** with $\theta$ detached: $\psi \leftarrow \psi - \eta_\psi \nabla_\psi(\lambda \mathcal{L}_{\text{RC}}(\psi))$
11: **end for**

---

Intuitively, $r$ measures the tracking gap between the generator output and the fake score's $x_0$-prediction.

We introduce a pair of dual potentials that induce opposite gradients w.r.t. $x_{\text{fake}}$ (NR for negative-residual, RC for residual-contraction):

$$\begin{aligned}
\mathcal{L}_{\text{NR}}(\theta) &:= -\tfrac{1}{2}\|r(x_0, x_t)\|^2, \\
\mathcal{L}_{\text{RC}}(\psi) &:= +\tfrac{1}{2}\|r(x_0, x_t)\|^2,
\end{aligned} \tag{18}$$

We use $\mathcal{L}_{\text{NR}}$ in the generator update and detach $\psi$. We use $\mathcal{L}_{\text{RC}}$ in the fake-score update and detach $\theta$.

Under the stop-gradient convention in Eq. (17), the other block is treated as constant, hence the two potentials induce opposite gradients:

$$\nabla_{x_{\text{fake}}}\mathcal{L}_{\text{NR}}(\theta) = r, \qquad \nabla_{x_{\text{fake}}}\mathcal{L}_{\text{RC}}(\psi) = -r. \tag{19}$$

Eq. (19) describes the dual effects in the $x_{\text{fake}}$-space. To connect to the generator behavior, we map them to $x_0$ through the dependence $x_{\text{fake}} = \mu_\psi(x_t, t)$ with $x_t = \alpha_t x_0 + \sigma_t \epsilon$. In particular, although $x_0$ is stop-gradient in $r = \text{sg}[x_0] - x_{\text{fake}}$, the generator still receives a non-zero gradient via the path $x_0 \to x_t \to x_{\text{fake}}$, yielding:

$$\begin{aligned}
\nabla_{x_0}\mathcal{L}_{\text{NR}} &= \left(\tfrac{\partial x_{\text{fake}}}{\partial x_0}\right)^* (x_0 - x_{\text{fake}}) \\
&= (\alpha_t J_\mu(x_t, t))^* (x_0 - x_{\text{fake}}),
\end{aligned} \tag{20}$$

where $J_\mu = \partial \mu_\psi/\partial x_t$. Under a tracking regime where $\alpha_t J_\mu(x_t, t) \approx P_t \succeq 0$ (approximately identity in the ideal case), the induced update direction on $x_0$ aligns with

$x_{\text{fake}} - x_0$, pulling the generator output toward the fake-score prediction and keeping the system close to score-consistency.

### 3.4.2. OVERALL OBJECTIVE AND TWO-STEP UPDATE

We retain the stable teacher stop-gradient Fisher objective in Eq. (13) as the distribution-matching objective and augment it with the outer correction:

$$\min_\theta \; \mathcal{L}_{\text{Fisher}}(\theta) + \lambda \mathcal{L}_{\text{NR}}(\theta), \tag{21}$$

while updating the fake score by residual contraction

$$\min_\psi \; \lambda \mathcal{L}_{\text{RC}}(\psi), \tag{22}$$

where $\lambda > 0$ balances distribution matching and tracking correction. In practice, we implement Eqs. (21)–(22) with a lightweight two-step (two-backward) procedure per iteration: first update $\theta$ using $\mathcal{L}_{\text{Fisher}} + \lambda \mathcal{L}_{\text{NR}}$ while treating the fake score as fixed, then update $\psi$ using $\lambda \mathcal{L}_{\text{RC}}$ while treating the generator as fixed. This yields a cooperative one-step bilevel update that explicitly controls tracking lag without computing second-order implicit-gradient terms, as visualized in Fig. 2c.

It is worth noting that the weight $\lambda$ should be moderate: too small $\lambda$ yields insufficient tracking correction, while too large $\lambda$ amplifies one-step lag (staleness) and makes the $x_{\text{fake}}$ update direction less accurate (Appendix A.2). Empirically, we find $\lambda = 0.1$ gives the best trade-off in our experiments.

Algorithm 1 summarizes the SGMD training procedure. We additionally provide PyTorch-style pseudocode in Appendix B to facilitate implementation.

### 3.4.3. COMPARISON WITH SIM

From the fake-score perspective, SGMD's asymptotically unbiased direction on $x_{\text{fake}}$ comes from two ingredients: (i) an unbiased generator update given by the teacher stop-gradient Fisher objective $\mathcal{L}_{\text{Fisher}}$ (Eq. (13)), and (ii) a dual pair of updates on the generator and the fake score induced by $\mathcal{L}_{\text{NR}}$ and $\mathcal{L}_{\text{RC}}$ (Eq. (18)). In contrast, although SIM also implicitly induces a dual structure, its generator update is not equivalent to the unbiased Fisher outer direction. Moreover, in SIM the relative strength between the two behaviors is fixed and cannot be tuned, whereas SGMD introduces an explicit weight $\lambda$ to control the tracking strength in Eqs. (21)–(22).

## 4. Experiments

To evaluate the efficacy of SGMD, we conduct experiments on the text-to-video (T2V) task. The SOTA base model Wan2.1-T2V-14B (Wang et al., 2025) is employed as the teacher. We follow a standard evaluation protocol for large-scale video diffusion distillation.

*Table 1.* Results on VBench-T2V under 4-step distillation. We report NFE as the total number of model forward evaluations. For the base model, we use classifier-free guidance (thus NFE is doubled). Abbreviations: NFE, number of function evaluations; Fake-R, fake-score updates per iteration; FVD, Fréchet Video Distance; OptFlow, optical-flow-based motion intensity; DynDeg, dynamic degree.

| Method | NFE↓ | Fake-R↓ | FVD↓ | OptFlow↑ | VBench-T2V | | |
| --- | --- | --- | --- | --- | --- | --- | --- |
| | | | | | Quality↑ | Semantic↑ | DynDeg↑ |
| Wan2.1-T2V-14B (base) | 100 | – | 0.0 | 9.41 | 86.67 | 84.44 | 94.26 |
| DMD2 (Yin et al., 2024a) | 4 | 5 | 115.1 | 4.51 | **85.05** | **77.46** | 80.56 |
| TSG-Fisher | 4 | 5 | 126.7 | 8.18 | 82.98 | 71.50 | **94.25** |
| TSG-SIM (Luo et al., 2024) | 4 | 1 | 193.0 | 3.27 | 82.68 | 73.21 | 59.72 |
| SGMD (ours) | 4 | 1 | **100.3** | 9.29 | 84.77 | 75.64 | 93.06 |

## 4.1. Experimental setup

All compared distillation methods (DMD2 (Yin et al., 2024a), TSG-Fisher, TSG-SIM (Luo et al., 2024), and SGMD) are trained using prompts only, without requiring paired ground-truth videos. We use a non-public prompt dataset (about 200K prompts) for training.

**Evaluation.** Large-scale video generation models (e.g., Wan2.1-T2V-14B (Wang et al., 2025)) exhibit strong motion dynamics and camera control, which is a key advantage over smaller models and a major factor behind their superior human preference. Since VBench (Huang et al., 2024b) alone may not be sufficiently sensitive to motion intensity, we use VBench as the standard benchmark (quality, semantic, and *dynamic degree*); we also compute the aggregated VBench total score for analysis. We additionally report an optical-flow-based motion intensity metric (Xu et al., 2023). This optical-flow metric is also adopted in Phased-DMD (Fan et al., 2025) for evaluating motion strength. Specifically, we compute per-frame optical-flow using UniMatch (Xu et al., 2023) and report the mean absolute flow magnitude averaged over frames and pixels. We also report FVD (Unterthiner et al., 2019) following the standard I3D-feature protocol. Unless otherwise specified, all distilled methods are evaluated using checkpoints after 300 training iterations, under the same inference and evaluation settings. We generate one video per evaluation instance at resolution $480 \times 832$ and duration 81 frames.

**Implementation.** We conduct experiments on 32 Nvidia-H100 GPUs, employing PyTorch FSDP (Zhao et al., 2023) and gradient checkpointing to reduce memory consumption. Context parallelism is applied for T2V distillation. The following settings are used consistently across all experiments: a batch size of 32; a fake diffusion model and a few-step generator initialized from the base model, with full-parameter training under a learning rate of $1 \times 10^{-6}$; AdamW optimizer (Loshchilov & Hutter, 2019) for both the fake diffusion and the generator, with hyperparameter $\beta_1 = 0$ and $\beta_2 = 0.999$. Euler solver is used in back-

*Table 2.* Ablations on $\lambda$ in SGMD.

| Setting | Total↑ | Quality↑ | Semantic↑ | DynDeg↑ |
| --- | --- | --- | --- | --- |
| $\lambda = 0.05$ | 81.92 | 83.68 | 74.90 | 93.55 |
| $\lambda = 0.1$ | **82.95** | **84.77** | **75.64** | 93.06 |
| $\lambda = 0.2$ | 82.01 | 84.06 | 73.81 | **94.23** |
| $\lambda = 0.5$ | 79.54 | 81.49 | 71.75 | 76.52 |

ward simulation due to its simplicity. All distillation experiments in this paper use 4-step sampling with timesteps $\{1000, 960, 889, 727\}$. For 4-step distillation, we adopt a self-forcing-style (Huang et al., 2025a) stochastic gradient truncation strategy to compute training gradients.

**Baselines.** We compare SGMD against three distilled baselines (DMD2, TSG-Fisher, and TSG-SIM), and additionally report the teacher base model as a reference. **Base model** refers to the original Wan2.1-T2V-14B sampler using 50 inference steps with classifier-free guidance (CFG) (Ho & Salimans, 2022). **DMD2** refers to DMD2 with a reverse-KL distribution matching objective using a fake-score tracker; we perform 5 fake-score updates per iteration and do not use any GAN loss (Goodfellow et al., 2014) for a controlled comparison. **TSG-Fisher** refers to optimizing the teacher stop-gradient Fisher objective as a distribution-matching baseline. **TSG-SIM** includes Score Implicit Matching under teacher stop-gradient control. We also include SGMD ablations in Sec. 4.3.

## 4.2. Results

**Quantitative results.** Table 1 summarizes the VBench-T2V, FVD, and optical-flow results under 4-step distillation. DMD2 achieves the strongest VBench quality and semantic scores among the distilled models, but it exhibits substantially weaker motion. In contrast, SGMD attains markedly stronger motion intensity (OptFlow) and dynamics (DynDeg) while remaining competitive on VBench quality and semantic scores, and it achieves the best FVD among the distilled models. We also observe that TSG-SIM does not improve motion-related metrics, likely because its effective update direction is reverse-KL-style and thus similar to

*(a)* SGMD (*Ours*)

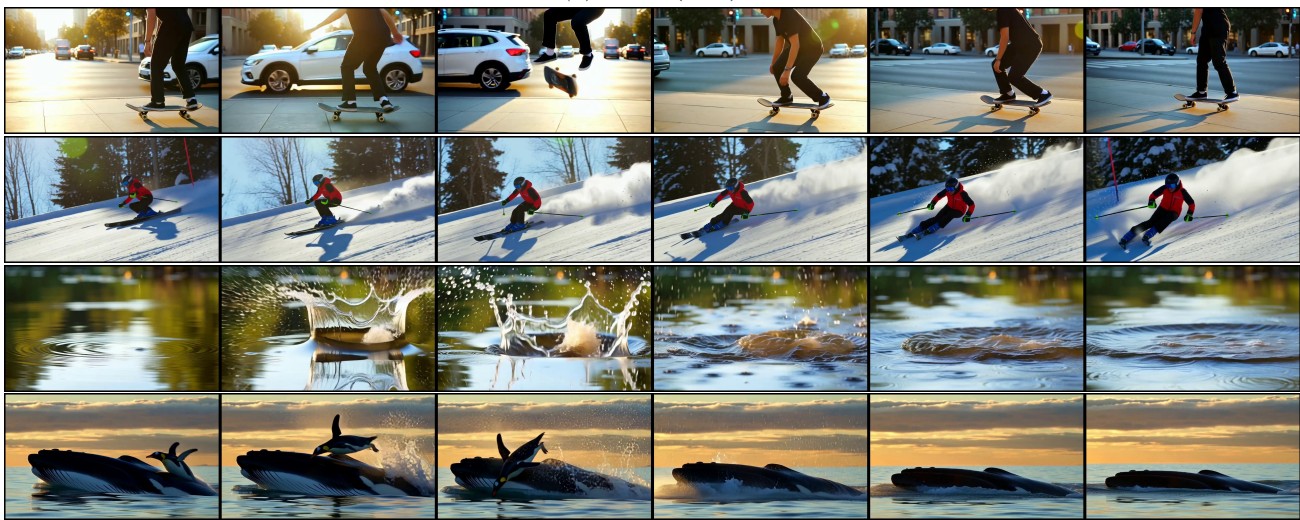

*(b)* DMD2 (Yin et al., 2024a)

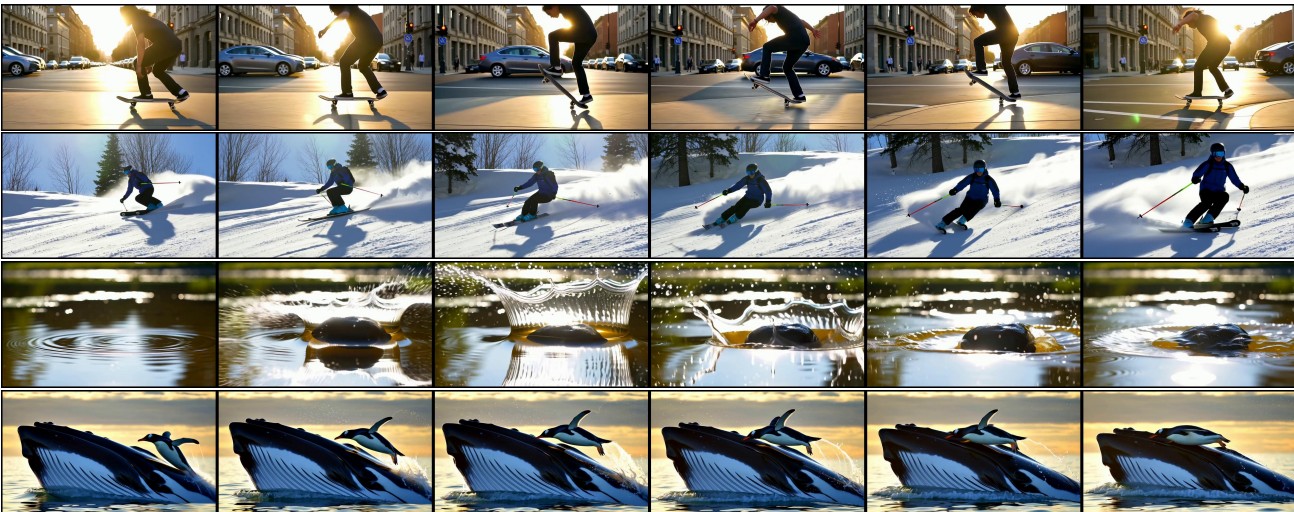

*Figure 3.* **Qualitative comparison (SGMD vs. DMD2).** Under comparable perceptual sharpness and visual quality, SGMD shows clearer temporal progression and larger motion changes across frames while maintaining good temporal consistency. For each 81-frame video, we show frames $\{0, 16, 32, 48, 64, 80\}$ as a preview.

*Table 3.* Human evaluation between SGMD and DMD2.

| Aspect | SGMD (%) | Tie (%) | DMD2 (%) |
|---|---|---|---|
| Overall Preference | 65 | 22 | 13 |
| Motion Quality | 71 | 5 | 24 |
| Text-Video Alignment | 4 | 90 | 6 |
| Visual Quality | 14 | 74 | 12 |

*Table 4.* VideoAlign evaluation between DMD2 and SGMD.

| Metric | DMD2 | SGMD |
|---|---|---|
| Overall | 18.86 | **19.36** |
| Visual Quality | **8.47** | 8.19 |
| Motion Quality | 4.15 | **4.99** |
| Text Align | **6.24** | 6.19 |

DMD2. Moreover, TSG-SIM implicitly imposes an overly strong dual term (roughly analogous to setting $\lambda = 1$ in SGMD), which can make the fake-score update direction inaccurate; consequently, TSG-SIM converges much more slowly and remains visibly blurry even after 400 training iterations.

**Human evaluation and VideoAlign.** To further validate the dynamics-quality trade-off from a perceptual perspective, we conduct a pairwise human evaluation between DMD2 and SGMD along four aspects: overall preference, motion quality, text-video alignment, and visual quality. As shown in Table 3, SGMD is preferred in overall preference (65%) and motion quality (71%), while text-video alignment and

visual quality are mostly judged as ties (90% and 74%, respectively). We additionally evaluate with VideoAlign (Liu et al., 2026), a human-feedback-trained reward model, as a scalable proxy for human preference. Table 4 shows a consistent trend: SGMD improves the overall score and motion quality, with small trade-offs in visual quality and text alignment. Together, these results indicate a favorable dynamics-quality trade-off: the motion gains are perceptually meaningful, while static visual quality and text alignment remain largely comparable.

**Mechanistic interpretation.** The motivating example in Fig. 1 helps interpret the trade-off in Table 1: reverse-KL objectives tend to strongly penalize allocating model mass in low-probability regions of the target distribution, leading to a more conservative update behavior. This offers an intuition for why reverse-KL-style distillation (e.g., DMD2) can preserve finer details yet suppress motion intensity, while Fisher-style objectives (e.g., Fisher/SGMD) provide smoother matching signals that more readily encourage stronger dynamics, potentially at the cost of a lower overall VBench score.

**Qualitative comparison.** Fig. 3 shows representative examples comparing SGMD and DMD2 under the same generation settings. We observe noticeably stronger motion dynamics from SGMD without an obvious loss of perceptual sharpness, while temporal consistency is well preserved. The prompts used in Fig. 3 are listed in Appendix E.

### 4.3. Analysis

**Ablation: objective components.** Table 1 highlights two complementary effects. (i) Compared to DMD2, the teacher stop-gradient Fisher objective (TSG-Fisher) substantially improves motion dynamics (OptFlow/DynDeg), at the cost of lower VBench quality and semantic scores. (ii) Compared to TSG-Fisher, SGMD's dual potentials (NR/RC) enable stable training with a reduced fake-score update ratio (Fake-R: $5 \rightarrow 1$), improving VBench quality and semantic scores while largely preserving strong dynamics.

**Ablation: $\lambda$ sensitivity.** We study the sensitivity of the tracking weight $\lambda$, which controls the trade-off between distribution matching and tracking correction in Eqs. (21)–(22). We sweep $\lambda \in \{0.05, 0.1, 0.2, 0.5\}$ while keeping the teacher, sampling steps, evaluation prompts, and optimization settings fixed. Table 2 shows that a moderate $\lambda$ yields the best trade-off: $\lambda = 0.1$ achieves the highest VBench total score, while $\lambda = 0.2$ gives the best DynDeg but incurs a lower total score and mild blur. A smaller $\lambda = 0.05$ preserves strong dynamics but tends to suffer from larger tracking lag, resulting in a lower total score. In contrast, a large $\lambda$ (e.g., $\lambda \geq 0.5$) often makes training difficult to converge, and the generated videos remain persistently blurry with poor details.

**Training efficiency.** By reducing the number of fake-score updates per iteration from 5 (DMD2-style baseline) to 1 (SGMD), we achieve an approximate $\sim 3\times$ training speedup every iteration compared to the DMD2-style baseline under our settings. The detailed counting and wall-clock estimate are provided in Appendix C.

## 5. Related Works

Our work builds on Distribution Matching Distillation (DMD) (Yin et al., 2024b), which couples a trainable generator with a student-side fake-score estimator under a pretrained teacher; related variants include DMD2 (Yin et al., 2024a) and Phased-DMD (Fan et al., 2025). Recent image-side work such as Flash-DMD (Chen et al., 2025) also studies efficient DMD-style distillation through timestep-aware training and joint reinforcement learning, which is complementary to our focus on fake-score tracking in video distillation. The closest related works are SIM (Luo et al., 2024) and SiD (Zhou et al., 2024), which address the two-timescale *tracking lag* between the fake score and the evolving generator by leveraging the fake-score-induced *implicit gradient*. SIM further characterizes the fixed relative weighting between the explicit and implicit terms, while SiD introduces an explicit reweighting coefficient. SGMD differs in two aspects. First, we adopt a *fake-score* perspective and analyze the net one-iteration update direction, which reveals a bias induced by SIM-style coupling. Second, we introduce a tunable pair of dual potentials (NR/RC) to decouple outer-loop correction from inner-loop contraction under the teacher stop-gradient Fisher objective. A more detailed comparison and derivations are provided in Appendix A.3.

## 6. Conclusion

In this paper, we propose **Score Gradient Matching Distillation (SGMD)** for few-step video diffusion distillation. SGMD adopts a fake-score perspective: it directly improves the fake score towards the teacher under a stable teacher stop-gradient Fisher objective, while updating the generator as a tracker to maintain score-consistency. This is realized by a pair of dual potentials that decouple outer-loop correction (NR) from inner-loop contraction (RC), yielding a lightweight two-step update per iteration. Empirically, SGMD improves motion dynamics and temporal consistency under 4-step distillation, and reduces training overhead by lowering the fake-score update ratio from 5 to 1, resulting in a $\sim 3\times$ speedup under our settings. A human study confirms that SGMD is preferred in motion quality and overall preference, while visual quality and text alignment remain comparable. We believe SGMD provides a principled and practical direction for stabilizing and accelerating few-step distillation of large-scale video diffusion models.

## Acknowledgements

This work was supported by the National Natural Science Foundation of China (Nos. 62525601, 62476018), and the Postdoctoral Fellowship Program of CPSF (No. BX20250487).

## Impact Statement

This work advances large-scale video generation by introducing a stable, few-step distillation framework. The societal implications align with typical considerations for generative ML: potential benefits in creative tools, simulation, and accessibility, alongside risks such as misuse and content authenticity, underscoring the importance of responsible release practices and evaluative transparency.

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

## A. Additional Proofs

### A.1. A formal justification of the fake-score perspective

We formalize the fake-score perspective without committing to any particular loss form. Let $s_{\text{fake}}(\cdot, t)$ be the learned fake score and $q_{\theta,t}$ be the generator-induced noisy-state distribution. Define the score-consistency set (a constraint manifold)

$$\mathcal{M} := \big\{ (\theta, \psi) : \ s_{\text{fake}}(x_t, t) = s_{q_{\theta,t}}(x_t) \text{ for } q_{\theta,t}\text{-a.e. } x_t \big\}, \tag{23}$$

and let $\mathcal{F}(\psi)$ denote an abstract teacher-alignment functional whose minimizer corresponds to matching the teacher score (*e.g.* a Fisher score-matching divergence under a teacher stop-gradient design). Then optimizing the fake score "as the objective" while using the generator to "track" can be viewed as minimizing $\mathcal{F}$ restricted to $\mathcal{M}$.

**Proposition A.1** (Constrained-view justification of the fake-score perspective)**.** *Assume $\mathcal{M}$ is non-empty and that for each $\psi$ in a neighborhood of interest there exists $\theta(\psi)$ such that $(\theta(\psi), \psi) \in \mathcal{M}$. Define the reduced objective $\widetilde{\mathcal{F}}(\psi) := \mathcal{F}(\psi)$ subject to $(\theta(\psi), \psi) \in \mathcal{M}$. Then any stationary point $\psi^\star$ of $\widetilde{\mathcal{F}}$ admits a compatible generator parameter $\theta^\star = \theta(\psi^\star)$ such that $(\theta^\star, \psi^\star) \in \mathcal{M}$, and the optimization can be interpreted as: (i) updating $\psi$ to decrease teacher mismatch, and (ii) updating $\theta$ to stay on (or close to) $\mathcal{M}$ so that $s_{\text{fake}}$ remains the score of the current generator-induced distribution.*

*Proof.* By assumption, for each $\psi$ in the neighborhood there exists at least one $\theta(\psi)$ such that $(\theta(\psi), \psi) \in \mathcal{M}$, hence the constrained problem defining $\widetilde{\mathcal{F}}$ is feasible. Let $\psi^\star$ be a stationary point of $\widetilde{\mathcal{F}}$ and define $\theta^\star := \theta(\psi^\star)$. Then $(\theta^\star, \psi^\star) \in \mathcal{M}$ by construction, which establishes the existence of a compatible generator parameter at $\psi^\star$. Moreover, interpreting the constrained optimization algorithmically yields the two roles: updates of $\psi$ target decreasing $\mathcal{F}$ (teacher alignment), while updates of $\theta$ act to restore feasibility (stay on or near $\mathcal{M}$), ensuring $s_{\text{fake}}$ remains compatible with the current generator-induced distribution. $\qquad\square$

### A.2. Why $\lambda$ should be moderate

We give a simple (local, stylized) two-timescale view that makes the trade-off in $\lambda$ explicit. Consider one coupled iteration $k$: the generator first updates $\theta$ (hence $x_0$ moves from $x_{0,k}$ to $x_{0,k+1}$), and then the fake score updates by residual contraction whose stop-gradient target is still $x_{0,k}$.

Let $r_k := x_{0,k} - x_{\text{fake},k}$ denote the tracking residual in the $x_{\text{fake}}$-space view (Eq. (19)). Approximating the inner update as a direct gradient step on $x_{\text{fake}}$, $x_{\text{fake},k+1} = x_{\text{fake},k} - \eta_\psi \lambda \nabla_{x_{\text{fake}}} \mathcal{L}_{\text{RC}} = x_{\text{fake},k} - \eta_\psi \lambda (x_{\text{fake},k} - x_{0,k})$, we obtain the linear recursion

$$r_{k+1} = (1 - \eta_\psi \lambda) r_k + \Delta x_{0,k}, \qquad \Delta x_{0,k} := x_{0,k+1} - x_{0,k}. \tag{24}$$

**Proposition A.2** (A simple $\lambda$ trade-off: stronger contraction but larger staleness)**.** *Assume $0 < \eta_\psi \lambda < 1$. Suppose the generator step satisfies a bound of the form*

$$\|\Delta x_{0,k}\| \le \eta_\theta (A + \lambda B) \qquad \text{for some constants } A, B \ge 0. \tag{25}$$

*Then: (i) Smaller $\lambda$ yields a looser asymptotic tracking-residual bound (scales as $A/\lambda$ when $A > 0$). (ii) Larger $\lambda$ yields a larger staleness bound (scales as $\lambda B$ when $B > 0$).*

*Proof.* **(i) Tracking-residual bound.** Let $\rho := 1 - \eta_\psi \lambda \in (0, 1)$. From Eq. (24) and the triangle inequality,

$$\|r_{k+1}\| \le \rho \|r_k\| + \|\Delta x_{0,k}\|. \tag{26}$$

Applying Eq. (25) gives $\|\Delta x_{0,k}\| \le M$ with $M := \eta_\theta (A + \lambda B)$, hence $\|r_{k+1}\| \le \rho \|r_k\| + M$. Unrolling the recursion yields $\|r_k\| \le \rho^k \|r_0\| + M \sum_{i=0}^{k-1} \rho^i = \rho^k \|r_0\| + \frac{M(1-\rho^k)}{1-\rho}$. Taking $\limsup$ and using $1 - \rho = \eta_\psi \lambda$ gives

$$\limsup_{k \to \infty} \|r_k\| \le \frac{\eta_\theta (A + \lambda B)}{\eta_\psi \lambda}. \tag{27}$$

**(ii) Staleness bound.** For staleness, by definition $g_k := x_{\text{fake},k} - \text{sg}[x_{0,k}]$ and $g_k^{\text{fresh}} := x_{\text{fake},k} - \text{sg}[x_{0,k+1}]$, hence $g_k - g_k^{\text{fresh}} = \text{sg}[x_{0,k+1} - x_{0,k}] = \Delta x_{0,k}$. Taking norms and using Eq. (25) yields

$$\|g_k - g_k^{\text{fresh}}\| = \|\Delta x_{0,k}\| \le \eta_\theta (A + \lambda B), \tag{28}$$

**Combining (i)–(ii).** Define a simple proxy of the coupled-iteration inaccuracy, $\mathcal{E}(\lambda) := \limsup_k \|r_k\| + \limsup_k \|g_k - g_k^{\text{fresh}}\|$. Substituting Eqs. (27)–(28) gives an upper bound of the form

$$\mathcal{E}(\lambda) \lesssim \underbrace{\frac{C_1}{\lambda}}_{\text{insufficient correction}} + \underbrace{C_2\lambda}_{\text{staleness increases}}, \tag{29}$$

for constants $C_1, C_2 > 0$ depending on $(\eta_\theta, \eta_\psi, A, B)$, which is minimized at a moderate $\lambda$. $\qquad\square$

### A.3. SiD vs. SGMD: a fake-score perspective

The main text already discusses SIM vs. SGMD under the fake-score perspective (Sec. 3.3). Here we focus on SiD (Zhou et al., 2024). Recall that SIM uses a fixed combination (Eq. (15)) $\mathcal{L}_{\text{SIM}}(\theta) = \mathcal{L}^{(1)}(\theta) + \mathcal{L}^{(2)}(\theta)$ (see also Eq. (16)), i.e., the ratio between the explicit term $\mathcal{L}^{(1)}$ and the implicit term $\mathcal{L}^{(2)}$ is fixed. In contrast, SiD can be viewed as reweighting the implicit term:

$$\mathcal{L}_{\text{SiD}}(\theta) = \mathcal{L}^{(1)}(\theta) + \alpha \mathcal{L}^{(2)}(\theta), \tag{30}$$

for a tunable scalar $\alpha$.

**Equivalent fake-score gradient for SiD.** Following the notation in Sec. 3.3, write $x_{\text{fake}} := \mu_{\text{fake}}(x_t, t)$ and $x_{\text{real}} := \mu_{\text{real}}(x_t, t)$. Using $\Delta_t = x_{\text{fake}} - x_{\text{real}}$ and $r_t = x_0 - x_{\text{fake}}$ (Eq. (16)), we have $\mathcal{L}_{\text{SiD}}(\theta) = c(t)\left(\frac{1}{2}\|\Delta_t\|^2 + \alpha\,\Delta_t^\top r_t\right)$ up to terms independent of $\theta$. Expanding in the $x$-prediction form gives

$$\mathcal{L}_{\text{SiD}}(\theta) = c(t)\Big(\tfrac{1}{2}\|x_{\text{real}}\|^2 + (\tfrac{1}{2} - \alpha)\|x_{\text{fake}}\|^2 + (\alpha - 1)\langle x_{\text{fake}}, x_{\text{real}}\rangle$$
$$+ \alpha\langle x_0, x_{\text{fake}}\rangle - \alpha\langle x_0, x_{\text{real}}\rangle\Big), \tag{31}$$

which reduces to Eq. (15) when $\alpha = 1$ (SIM). Taking the total derivative w.r.t. $x_{\text{fake}}$ (including the $\theta$-induced dependence $x_0 = x_0(x_{\text{fake}})$) yields the effective direction:

$$\nabla_{x_{\text{fake}}}\mathcal{L}_{\text{SiD}} = c(t)\Big(\alpha(x_0 - x_{\text{fake}}) + (1 - \alpha)(x_{\text{fake}} - x_{\text{real}}) + \alpha\left(\tfrac{dx_0}{dx_{\text{fake}}}\right)^*(x_{\text{fake}} - x_{\text{real}})\Big), \tag{32}$$

where $(\cdot)^*$ denotes the adjoint (VJP) operator under the Frobenius inner product, consistent with Eq. (16). Eq. (32) makes explicit that changing $\alpha$ not only rescales the SIM-style tracking term, but also introduces an additional bias term proportional to $(1 - \alpha)(x_{\text{fake}} - x_{\text{real}})$ in the fake-score view, which is absent when $\alpha = 1$.

## B. PyTorch-style pseudocode

We provide PyTorch-style pseudocode for SGMD to clarify the stop-gradient design and the two-backward update per iteration.

```
for batch in dataloader:
    # sample noise level
    t = sample_t(batch_size)
    alpha_t, sigma_t = 1 - t, t

    # generator forward
    x0 = G_theta(batch)   # may include conditioning (e.g., text/image)
    eps = torch.randn_like(x0)
    x_t = alpha_t * x0 + t * eps

    # forward passes (shared for both updates)
    x_fake = fake_score(x_t, t)
    with torch.no_grad():
        x_real_cond = teacher(x_t.detach(), t, cond)
        x_real_uncond = teacher(x_t.detach(), t, uncond)
        x_real = x_real_cond + cfg_scale * (x_real_cond - x_real_uncond)
```

```
delta = x_fake - x_real                                    19
c = alpha_t**2 / sigma_t**4                                20
L_fisher = 0.5 * (c * (delta**2)).mean()                   21
                                                           22
# residual uses stop-grad on x0                            23
r = x0.detach() - x_fake                                   24
L_NR = -0.5 * (r**2).mean()                                25
L_RC =  0.5 * (r**2).mean()                                26
                                                           27
# 1) update generator                                      28
opt_G.zero_grad()                                          29
(L_fisher + lamb * L_NR).backward()                        30
opt_G.step()                                               31
                                                           32
# 2) update fake-score                                     33
opt_F.zero_grad()                                          34
(lamb * L_RC).backward()                                   35
opt_F.step()                                               36
```

## C. Training-time breakdown

We provide a simple operator-count and wall-clock estimate to compare SGMD with a DMD2-style baseline under our settings. We use a representative setting where the DMD2-style baseline performs one generator update and $K = 5$ fake-score updates per iteration. For 4-step distillation, we approximate the backward simulation cost as 2.5 forward evaluations (due to solver unrolling).

**Operator count.** **SGMD** uses an expected $F_{\text{SGMD}} \approx 6.5$ forward evaluations per iteration and performs one short-range backward and one long-range backward. **DMD2-style baseline** uses $F_{\text{baseline}} \approx 5.5(1 + K) = 33$ forward evaluations per iteration and performs $B_{\text{baseline}}^{\text{short}} = 1 + K = 6$ short-range backwards (no long-range backward).

**Wall-clock estimate.** Under our training settings, one forward evaluation takes $\approx 5$s, one short-range backward takes $\approx 15$s, and one long-range backward takes $\approx 30$s. Therefore,

$$T_{\text{SGMD}} \approx 6.5 \times 5 + 1 \times 30 + 1 \times 15 = 77.5 \text{ s}, \qquad T_{\text{baseline}} \approx 33 \times 5 + 6 \times 15 = 255 \text{ s},$$

yielding an overall speedup of $T_{\text{baseline}}/T_{\text{SGMD}} \approx 3.3\times$ (about $\sim 3\times$ in practice).

## D. 1D mixture fitting details: reverse-KL vs. Fisher

We design a simple 1D mixture-fitting problem to qualitatively contrast reverse-KL and Fisher divergence matching and to illustrate why reverse-KL-style updates can be more conservative, while Fisher-style updates provide smoother matching signals.

**Target distribution.** We fix an asymmetric two-component Gaussian mixture

$$p(x) = 0.75\,\mathcal{N}(x; -1.2, 0.55^2) + 0.25\,\mathcal{N}(x; 2.0, 0.85^2).$$

**Model family.** We use a symmetric two-component mixture with equal weights,

$$q_\phi(x) = \tfrac{1}{2}\mathcal{N}(x; +m, s^2) + \tfrac{1}{2}\mathcal{N}(x; -m, s^2),$$

where $\phi = (m, s)$ with $m \geq 0$ and $s > 0$.

**Objectives.** We compare (i) reverse-KL ($\text{KL}(q\|p)$),

$$\text{KL}(q\|p) = \int q(x) \log \frac{q(x)}{p(x)}\, dx,$$

and (ii) Fisher divergence,

$$\mathcal{F}(q, p) = \int q(x) \left(\partial_x \log q(x) - \partial_x \log p(x)\right)^2 dx.$$

Reverse-KL ($\mathrm{KL}(q\|p)$) assigns a large penalty when $q$ places non-trivial mass in regions where $p(x)$ is small; this tends to discourage exploratory mass in low-probability regions and leads to more conservative updates. In contrast, the Fisher divergence emphasizes matching local score fields and typically yields a smoother, more global correction signal.

**Numerical approximation and optimization.** All integrals are approximated by numerical quadrature on a fixed uniform grid $x \in [-7, 7]$ with 4001 points. We optimize $\phi$ by gradient-based updates (Adam) for 2500 steps with learning rate $5 \times 10^{-2}$, starting from a common initialization $(m, s) = (1.0, 1.2)$. Fig. 1 visualizes the fitted $q_\phi$ under the two objectives against the fixed target $p$.

## E. Qualitative prompts

We list the prompts used for the qualitative comparison in Fig. 3.

| Video | Prompt |
|---|---|
| 1 | Tracking shot, daytime, sunlight, medium wide shot, side lighting, warm colors. A skateboarder glides down a bustling city street, performing smooth tricks amidst the urban landscape. The sun casts a golden glow, highlighting the skateboarder's agile movements. Cars and pedestrians pass by in the background, adding to the dynamic energy of the scene. The skateboarder's hair flows freely as they execute a series of jumps and spins, showcasing their skill and grace. |
| 2 | Day time, sunny lighting, side lighting, wide shot, center composition. A skier accelerates down a steep slope during a downhill race. The sun casts sharp shadows on the snow-covered mountainside, highlighting the skier's determined expression and sleek skiing gear. Snowflakes drift in the cold air, adding a touch of movement to the scene. Trees line the edge of the slope, their branches swaying slightly in the breeze. The skier leans forward, arms extended for balance, as they carve through the powder snow at high speed. |
| 3 | A high-speed video of a water-filled balloon being sliced open, with water flowing out in a controlled manner. The scene is set during daytime with sunlight streaming through a window, casting soft lighting. The camera captures a medium shot, focusing on the precise moment the balloon is cut, revealing droplets of water glistening as they fall in slow motion. In the background, leaves rustle gently in the breeze, adding movement to the scene. |
| 4 | In a stunning moment captured at sunrise time, a penguin flies dramatically into the mouth of a blue whale as it breaks through the surface of the ocean. Soft sunlight bathes the scene in warm hues, casting gentle side lighting that highlights the spray of water and the sleek bodies of both animals. The camera captures this extraordinary event from a medium wide shot, emphasizing the vastness of the ocean and the remarkable scale of the interaction. Clouds drift lazily across the sky, adding movement to the serene backdrop. |

*Table 5.* Prompts corresponding to the four videos in Fig. 3 (in order).

