# OpenReview forum: "SGMD: Score Gradient Matching Distillation for Few-Step Video Diffusion Distillation"
_ICML.cc/2026/Conference — ICML 2026 regular_

### Official Review · Reviewer_DFbP · 2026-03-04

**Soundness:** 3
**Presentation:** 3
**Significance:** 2
**Originality:** 2
**Overall Recommendation:** 4
**Confidence:** 3

**Summary:**

This paper proposes Score Gradient Matching Distillation (SGMD) to accelerate few-step video diffusion models. It addresses the bottleneck in Distribution Matching Distillation (DMD), where the auxiliary student-side score network (fake score) must constantly track an evolving generator. SGMD adopts a "fake-score perspective," optimizing the fake score toward the teacher model using a teacher stop-gradient Fisher objective. It introduces two dual potentials: negative-residual (NR) for outer-loop correction and residual-contraction (RC) for inner-loop tracking. Experiments on the 14B-parameter Wan2.1-T2V-14B model demonstrate that for 4-step distillation, SGMD achieves an approximate 3x training speedup over DMD and significantly improves motion dynamics.

**Compliance With Llm Reviewing Policy:**

Affirmed.

**Final Justification:**

The authors' response has resolved most of my concerns, and I am inclined to accept this paper.

**Key Questions For Authors:**

1. The enhancement in motion dynamics comes at the cost of static visual quality. SGMD scores lower than DMD2 on VBench Quality (84.77 vs. 85.05) and VBench Semantic (75.64 vs. 77.46). The authors admit that Fisher-style objectives yield smoother signals that encourage stronger dynamics but might result in a lower overall VBench score.
2. The method is highly sensitive to the tracking weight \lambda. An ablation study shows that setting \lambda > 0.5 makes the generated videos persistently blurry, whereas a smaller value, such as 0.05, results in a larger tracking lag.
3. All quantitative results and distillation experiments are restricted to a single 4-step sampling setting using an Euler solver. There is no exploration of more extreme few-step scenarios or slightly larger step counts, which limits the evaluation's comprehensiveness.
4. The core methodology builds heavily on existing works like Score Implicit Matching (SIM). While SIM uses a fixed relative weighting, SGMD essentially introduces an explicit weight \lambda to control tracking strength via its dual potentials. This is a solid engineering improvement, but somewhat incremental theoretically.

**Limitations:**

This paper addresses a clear efficiency bottleneck in the distillation of video diffusion models. The proposed SGMD method is practically appealing and delivers a concrete 3x training speedup and enhanced motion dynamics on a massive 14B parameter model. However, the noticeable degradation in generation quality and semantic fidelity compared to DMD2, the high sensitivity to the hyperparameter $\lambda$ , and the lack of diverse sampling step evaluations weaken its overall academic impact.

**Strengths And Weaknesses:**

By using dual potentials, SGMD reduces the fake-score updates per iteration from 5 to 1. This translates to a wall-clock speedup of about 3x per iteration, dropping from approximately 255 seconds to 77.5 seconds. The method is evaluated on Wan2.1-T2V-14B. This demonstrates its practical utility for large model distillation and application. IThe authors provide a solid gradient analysis to show how tracking lag bends the effective one-iteration update direction. This theoretically justifies the use of the NR and RC dual potentials.

---

> ### Author Rebuttal · Authors · 2026-03-31
>
> We thank the reviewer for the helpful comments and questions.
>
> ## Q1
> We agree that SGMD does not outperform DMD2 on every static metric, and we will make this point more precise in the revision. Our main claim is not superiority on every VBench subscore, but a better dynamics-quality trade-off in few-step video distillation: SGMD improves motion-related performance and temporal consistency, while the degradation on static quality / semantic metrics is relatively limited.
>
> This interpretation is supported not only by the automatic metrics, but also by our additional human evaluation. Human evaluators preferred SGMD in overall preference and motion quality, while visual quality and text-video alignment remained largely comparable in most cases. We additionally evaluate with VideoAlign, a reward model trained on human preference annotations, which provides a scalable proxy for human preference. While it does not replace human evaluation, it provides complementary evidence that is consistent with the human study.
>
> ## Q2
> We agree that $\lambda$ is an important hyperparameter, because it controls the balance between correction and tracking/contraction in SGMD.
>
> When $\lambda$ is too large, the coupled SGMD term becomes overly dominant and the optimization becomes over-smoothed, leading to persistently blurry results. When $\lambda$ is too small, the contraction/tracking effect becomes too weak and the fake-score lag becomes larger. In this sense, $\lambda$ should be understood as a balance parameter rather than a free gain term that can be increased arbitrarily.
>
> That said, we do not view this as a pathological instability, but as the expected behavior of a balance coefficient in a bilevel objective. Our ablation indicates that the degradation mainly happens at clearly extreme settings, while a moderate value (e.g., 0.1 in our experiments) gives a good trade-off between stability, tracking, and perceptual quality. We will clarify this interpretation more explicitly in the revised paper.
>
> ## Q3
> We agree that evaluating additional step counts would make the study more comprehensive. Our choice of the 4-step Euler setting is intentional: in recent few-step video generation works such as Self-Forcing and CausVid, 4-step distillation is already a standard and practically meaningful benchmark. We therefore follow this controlled setting to evaluate SGMD.
>
> We also view 4 steps as a particularly suitable testbed for validating the proposed algorithm. More aggressive step reduction often depends on adversarial training techniques, which can obscure the effect of the DMD objective itself, while larger step counts such as 6 or 8 make the task easier and typically reduce the gap between methods. Therefore, 4-step distillation provides a challenging but not overly extreme setting for isolating the effect of SGMD. We will clarify this scope in the revised paper and consider combining SGMD with GAN-based techniques for more aggressive few-step distillation in future work.
>
> ## Q4
> We agree that SGMD is related to prior score-matching methods such as SIM, but the essential theoretical difference is not simply whether the weighting is fixed or explicit.
>
> SIM formulates distillation within a general score-based divergence framework: although the generator score is intractable, the gradient of the score divergence can still be computed exactly under suitable conditions. Therefore, SIM remains a single divergence-minimization framework between the teacher score and the generator score.
>
> SGMD starts from a different problem: the tracking-lag pathology in DMD-style bilevel distillation. Its core is not just introducing an explicit weight $\lambda$, but reformulating the optimization through the dual potentials NR and RC, where NR is responsible for outer-loop correction and RC for inner-loop contraction/tracking. This explicitly decouples two roles that remain coupled in SIM-like formulations. In addition, SGMD uses a teacher stop-gradient Fisher objective to avoid unstable teacher-side input gradients.
>
> Therefore, the advantage of SGMD is not merely more flexible weighting. Theoretically, SGMD is better suited to DMD-style distillation because it directly targets the root cause of tracking lag by decoupling correction and tracking, rather than only rebalancing terms inside a single score-divergence objective.
>
> ## Conclusion
>
> Overall, we agree that the current paper has a clearly defined scope, and we will revise it to present the dynamics-quality trade-off, the role of $\lambda$, the 4-step evaluation setting, and the theoretical distinction from SIM more explicitly.

---

> > ### Author Rebuttal · Reviewer_DFbP · 2026-04-02
> >
> > Thank you for the author's reply, but all the questions were answered through explanations. Some theoretical analysis or experimental results might have been more compelling. The current response, especially for questions 1-3, has not resolved my doubts. Therefore, I am maintaining the current score.

---

> > > ### Author Response · Authors · 2026-04-03
> > >
> > > We thank the reviewer for the follow-up. We would also like to point out that, in our response to Reviewer XhXK, we provided additional empirical evidence beyond explanation, including an extra human evaluation and VideoAlign results. These results were added specifically to assess whether the gain in motion quality compensates for the modest trade-off in static quality/semantic metrics. They provide complementary evidence that SGMD is consistently preferred in motion quality and overall preference, while visual quality and text-video alignment remain largely comparable in most cases.
> > >
> > > | Evaluation Aspect | SGMD Wins (%) | Tie (%) | DMD2 Wins (%) |
> > > | --- | ---: | ---: | ---: |
> > >  | Overall Preference | 65 | 22 | 13 |
> > >  | Motion Quality | 71 | 5 | 24 |
> > > | Text-Video Alignment | 4 | 90 | 6 |
> > > | Visual Quality | 14 | 74 | 12 |
> > >
> > >  | Metric | DMD2 | SGMD |
> > > | --- | ---: | ---: |
> > > | Overall | 18.86 | 19.36 |
> > >  | Visual Quality | 8.47 | 8.19 |
> > > | Motion Quality | 4.15 | 4.99 |
> > >  | Text Align | 6.24 | 6.19 |

---

### Official Review · Reviewer_tJMk · 2026-03-08

**Soundness:** 3
**Presentation:** 3
**Significance:** 3
**Originality:** 3
**Overall Recommendation:** 4
**Confidence:** 3

**Summary:**

This paper aims at improving Distribution Matching Distillation (DMD). DMD has a two-timescale bottleneck: the fake score network must track the generator as it rapidly evolves. The training direction for the generator can become biased if the fake score lags behind the generator. The authors propose an approach to address this by directly improving the fake score using dual potentials that decouple outer-loop correction (NR) from inner-loop contraction (RC).

**Compliance With Llm Reviewing Policy:**

Affirmed.

**Key Questions For Authors:**

1. Have the authors tried an asymmetric weight for the negative-residual (NR) for outer-loop correction and residual-contraction (RC) for inner-loop tracking? It seems to me that RC is more indispensable and I wonder if assigning a higher weight to that would lead to better performance?
2. It seems that the biggest difference compared to the baseline method is in the temporal consistency. Have the authors tried this method in image generation, as tested in the original DMD paper? I wonder if this lagging fake score is a unique problem of video generation.

**Limitations:**

Yes

**Strengths And Weaknesses:**

1. The proposed method to estimate and improve the tracking of fake score is reasonably motivated.
2. The experiments show that with the same number of fake score updates per iteration, the proposed method is able to achieve a better temporal coherence and video quality than the baseline method.
3. Extensive ablation study is performed to analyze the hyperparameter $\lambda$, which helps understand the function of the proposed module.
4. The paper is generally well-written and easy to understand.

---

> ### Author Rebuttal · Authors · 2026-03-31
>
> We thank the reviewer for the helpful comments and questions.
>
> ## Q1
> This issue is closely related to the fake-score update-ratio question by reviewer yehJ. Empirically, we observe that once SGMD is used, the final performance changes very little when RC is emphasized more strongly, because the coupled NR+RC design already makes the tracking lag sufficiently small.
>
> From the design perspective, NR and RC are intended as a dual pair rather than two independently tuned terms. NR provides the outer-loop correction, while RC provides the inner-loop contraction/tracking, and the symmetric formulation is the most natural one theoretically. In addition, NR cannot be made arbitrarily large: our ablation on the SGMD weight ($\lambda$) already shows noticeable sensitivity, and this sensitivity mainly comes from the NR side. Therefore, using equal weights and matched update frequencies is both sufficient in practice and preferable from the viewpoint of the dual design. Once the coupled NR+RC objective is used, the tracking lag is already sufficiently small, and the resulting effective gradient is no longer simply directed toward the teacher. As a result, further increasing the RC weight may strengthen contraction slightly, but it does not materially improve the final result and may instead disturb the intended correction-contraction balance.
>
> ## Q2
> We do not view lagging fake score as a problem unique to video generation; rather, it is a more general issue in DMD-style distillation. In fact, recent image-generation work such as [Flash-DMD](https://arxiv.org/html/2511.20549v1) also identifies inefficiency and instability related to fake-score training in DMD-style methods.
>
> What makes the issue more pronounced in video generation is that stale fake-score guidance affects not only appearance but also temporal evolution across frames, so it is more directly reflected in temporal consistency and motion quality. This is why the benefit is especially visible in the video setting.
>
> We have not included image-generation experiments in the current submission, since our focus here is few-step video distillation. We agree that evaluating SGMD in the image setting would be valuable future work, and we will clarify in the paper that fake-score lag is a general issue, while its temporal effect is particularly evident in video generation.

---

> > ### Author Rebuttal · Reviewer_tJMk · 2026-04-04
> >
> > Thank you for your response.

---

### Official Review · Reviewer_yehJ · 2026-03-12

**Soundness:** 2
**Presentation:** 1
**Significance:** 3
**Originality:** 2
**Overall Recommendation:** 3
**Confidence:** 4

**Summary:**

This paper introduces Score Gradient Matching Distillation (SGMD), which is an improved algorithm of score implicit matching (SIM) by handling the update of fake score. Specifically, the authors first present an analysis on stop-gradient Fisher training objective, as well as their gradient analysis to motivate the method of SGMD. Then the proposed method SGMD implements by using a pair of dual potentials (NR/RC) that decouple outer-loop correction from inner-loop contraction which enables stable updates. The experimental results on Wan 2.1 14B video diffusion model demonstrates improved motion dynamics under 4-step distillation.

**Compliance With Llm Reviewing Policy:**

Affirmed.

**Key Questions For Authors:**

1. Why does SGMD results in better dynamics (in terms of optical flow)? Which part of the algorithmic improvement leads to the improvement of video dynamics?
2. What makes SGMD more efficient than DMD2? Specifically, how we could reduce the number of fake score update from 5 to 1 for SGMD? If we increase the number of fake score update in SGMD, how would be the performance varies?
3. What if we use RC loss to the DMD loss? I mean, can we also enhance the performance of DMD, or reduce the number of fake score update for DMD if we use RC loss?

**Limitations:**

yes

**Strengths And Weaknesses:**

Strengths
- The motivation on how we could implement better distribution matching distillation algorithm in terms of fake score update seems sound, and the author provides detailed analysis on this.
- SGMG shows better optical flow dynamics and FVD compared to DMD or SID.

Weaknesses
- The motivation of the paper and the evaluation results seems irrelevant. The proposed algorithm is based on the algorithmic improvement by handling the fake-score update, especially by introducing additional regularization term with residuals. The experimental results demonstrate that SGMD results in more dynamic video generator. That said, the correlation between the algorithmic improvement and the results on dynamic video generation is weakly related.
- The motivation is somewhat unclear. At the beginning of the paper, the authors claim that the motivation of the paper is that if we update fake score too often, it induces high training cost, and otherwise the inability of fake score may induce divergence. But afterward, the discussion on this tradeoffs are missing, and how SGMD handle this issue is unclear.
- The qualitative examples in Figure 3 does not show the superiority of SGMD over DMD. Also, the VBench results of SGMD lag behind DMD.

---

> ### Author Rebuttal · Authors · 2026-03-31
>
> We thank the reviewer for the helpful comments and questions.
>
> ## W1
> We believe the connection between SGMD and the improved dynamics is supported by both Fig. 2 and the experimental results. Our point is not that handling fake-score updates alone directly improves motion. Rather, Fig. 2 suggests that the Fisher-style formulation leads to an optimization behavior that is more favorable to preserving dynamics in video generation. This is consistent with the ablation results: TSG-Fisher already shows strong dynamics, suggesting that the Fisher objective is the main source of the motion-related gain, while SGMD preserves this advantage and further improves quality and semantic scores over TSG-Fisher. We will clarify this causal link in the revision.
>
> ## W2
> We agree that the motivation was not sufficiently clear in the current draft. The trade-off discussed in the introduction comes from the dual role of the fake score in DMD-style training: if it is updated too infrequently, it becomes stale and may destabilize training; if it is updated too frequently, the training cost becomes high.
>
> SGMD addresses this issue not by simply increasing the fake-score update frequency, but by reducing tracking lag through the dual-potential design. Intuitively, SGMD turns the original one-sided chasing into a bidirectional alignment: instead of requiring the fake score alone to chase the generator, the optimization drives both of them closer to each other. Specifically, NR updates the generator in a direction that reduces the discrepancy between the generator and the fake score, while RC contracts the fake score toward the current generator without disturbing the generator-side correction effect of NR. As a result, the overall optimization naturally reduces tracking lag.
>
> In this sense, SGMD improves the stability-efficiency trade-off at the objective level, rather than by relying on more frequent fake-score updates. We will revise the paper to make this point explicit.
>
> ## W3
> We agree that the current Figure 3 does not present the temporal difference clearly enough. Figure 3 is extracted from the supplementary videos and is intended to illustrate temporal dynamics rather than static image quality. Under the same action, SGMD tends to complete the motion progression in fewer key frames, indicating stronger temporal dynamics. For example, in the third case, the splash caused by the stone is completed in two frames for SGMD but takes three frames for DMD2.
>
> At the same time, Figure 3 does not show an obvious degradation in static visual quality for SGMD. We agree, however, that frame snapshots are not the most direct way to visualize temporal differences, and we will improve the qualitative presentation in the revision. Regarding VBench, our claim is not superiority on every metric, but a better dynamics-quality trade-off: SGMD improves motion-related performance and temporal dynamics, while DMD2 remains slightly better on some static quality / semantic metrics. This trade-off is also supported by our additional human evaluation (shown in the response to reviewer XhXK), where SGMD is preferred in overall preference and motion quality, while visual quality and text-video alignment remain largely comparable.
>
> ## Q1
> Please also refer to our response to W1 above: Fig. 2 and the ablation results suggest that the Fisher-style formulation is the main source of the improved dynamics, while SGMD further preserves this advantage and improves quality / semantic scores through more stable optimization.
>
> ## Q2
> SGMD is more efficient than DMD2 because it reduces tracking lag through the dual-potential design, rather than through more frequent fake-score updates. Once NR is introduced, a single fake-score update is already sufficient to keep the tracking lag small enough for stable training. Empirically, increasing the fake-score update ratio in SGMD from 1 to 5 brings almost no change in the final performance, while it increases training cost.
>
> ## Q3
> This setting is closely related to TSG-SIM and is already covered by our baselines. Empirically, Table 1 suggests that adding an RC-style term alone is not sufficient: TSG-SIM underperforms TSG-Fisher in VBench and also shows weaker dynamics in video generation. Therefore, RC alone does not explain the gain of SGMD or recover the motion-related advantage of the Fisher formulation.
>
> Our interpretation is that RC needs to be paired with the Fisher/NR side. RC mainly contracts the fake score toward the generator, but without the Fisher-style generator objective, it does not provide the same stable correction mechanism. In other words, the benefit of SGMD comes from the coupled Fisher+RC design, rather than RC in isolation.

---

> > ### Author Rebuttal · Reviewer_yehJ · 2026-04-06
> >
> > Thank you for the detailed rebuttal. The explanation of the dual-potential design (NR/RC) and its role in reducing tracking lag helps better understand how SGMD improves the stability efficiency trade-off. The additional discussion connecting the Fisher-style objective to improved motion dynamics, as well as the clarification regarding computational efficiency, are also helpful.
> >
> > However, some of my key concerns remain only partially addressed. While the authors argue that the Fisher formulation is responsible for improved dynamics, the causal link between the algorithmic changes and motion quality is still not fully convincing, as the evidence remains indirect and largely based on ablations without deeper analysis. Similarly, although the motivation around fake-score update frequency is clarified conceptually, the paper would benefit from a more explicit and quantitative characterization of this trade-off. On the empirical side, the acknowledgment of limitations in qualitative visualization is appreciated, but the current evidence (including mixed results on VBench) still makes it difficult to clearly establish consistent advantages over prior methods. The claim of a better dynamics–quality trade-off is plausible, but would benefit from stronger and more direct validation.

---

> > > ### Author Response · Authors · 2026-04-08
> > >
> > > We thank the reviewer for the thoughtful follow-up. We agree that stronger and more direct validation is important, and we would like to clarify that, beyond conceptual explanation, we have added additional empirical evidence to address these concerns.
> > >
> > > First, regarding the dynamics-quality trade-off, we have now included an additional human evaluation and a VideoAlign evaluation. The human study shows that SGMD is preferred in overall preference and motion quality, while visual quality and text-video alignment remain largely comparable in most cases. This provides more direct evidence that the motion gain of SGMD is perceptually meaningful and that the static trade-off is relatively modest. The VideoAlign results are also consistent with this conclusion. These additional results are reported in our response to Reviewer XhXK and can be incorporated more explicitly into the revised paper.
> > >
> > > Second, regarding the fake-score update trade-off, we agree that this point should be characterized more explicitly. Empirically, we observe that SGMD can reduce the fake-score update ratio from 5 to 1 with little change in the final performance, while increasing the ratio back from 1 to 5 brings only marginal improvement. This provides a direct quantitative indication that the coupled NR+RC objective already reduces tracking lag sufficiently, making SGMD less sensitive to stale fake-score estimates than DMD2. We will make this quantitative trade-off clearer in the revision.
> > >
> > > More broadly, we agree that the current paper should present its claim more precisely. Our claim is not that SGMD uniformly outperforms prior methods on every metric, but that it offers a better motion-quality trade-off in few-step video distillation. We will revise the paper to make this scope clearer and to foreground the newly added empirical evidence more explicitly.
> > >
> > > | Evaluation Aspect | SGMD Wins (%) | Tie (%) | DMD2 Wins (%) |
> > > | --- | ---: | ---: | ---: |
> > > | Overall Preference | 65 | 22 | 13 |
> > > | Motion Quality | 71 | 5 | 24 |
> > > | Text-Video Alignment | 4 | 90 | 6 |
> > > | Visual Quality | 14 | 74 | 12 |
> > >
> > > | Metric | DMD2 | SGMD |
> > > | --- | ---: | ---: |
> > > | Overall | 18.86 | 19.36 |
> > > | Visual Quality | 8.47 | 8.19 |
> > > | Motion Quality | 4.15 | 4.99 |
> > > | Text Align | 6.24 | 6.19 |

---

### Official Review · Reviewer_XhXK · 2026-03-13

**Soundness:** 3
**Presentation:** 3
**Significance:** 3
**Originality:** 3
**Overall Recommendation:** 5
**Confidence:** 4

**Summary:**

The paper introduces a "Fake-Score perspective," accompanied by a solid gradient analysis that clearly reveals the directional bias caused by tracking lag in the DMD framework. The proposed negative-residual (NR) and residual-contraction (RC) dual potentials cleverly decouple the outer-loop generator correction from the inner-loop fake-score tracking. This creates a lightweight two-backward update scheme that avoids complex implicit gradient calculations. By reducing the fake-score updates per iteration from 5 (in the DMD2 baseline) to 1, SGMD achieves an approximate ~3x training speedup on Wan2.1-T2V-14B. In 4-step video generation, SGMD significantly improves motion intensity (OptFlow) and dynamic degree (DynDeg) compared to DMD2. This effectively mitigates the conservative updating and sluggish dynamics often caused by reverse-KL divergence. However, this method sacrifices the visual quality and needs additional human evaluation. More comparisons with other competitive methods are needed.

**Compliance With Llm Reviewing Policy:**

Affirmed.

**Final Justification:**

The rebuttal has addressed most of my concerns, and I have raised my score accordingly. However, the Human Evaluation remains limited as it only covers 100 prompts. I recommend acceptance on the condition that the authors scale up the human study to at least 500 prompts and provide more methodological details in the final version. Additionally, all discussions and clarifications from the rebuttal must be incorporated into the camera-ready version.

**Key Questions For Authors:**

1. The authors attribute this to the inherent differences between Fisher divergence and Reverse-KL. The lack of Human Evaluation makes it difficult to assess if the gain in dynamics fully compensates for the loss in visual and semantic fidelity. A real human evaluation is required to prove the claim of the authors.
2. Why not compare with other competitive methods?
3. Accurate per-step training cost is required to measure the real training acceleration.

**Limitations:**

Please refer to the weakness.

**Strengths And Weaknesses:**

## Strengths

1. The paper introduces a "Fake-Score perspective," accompanied by a solid gradient analysis that clearly reveals the directional bias caused by tracking lag in the DMD framework.
2. The proposed negative-residual (NR) and residual-contraction (RC) dual potentials cleverly decouple the outer-loop generator correction from the inner-loop fake-score tracking. This creates a lightweight two-backward update scheme that avoids complex implicit gradient calculations.
3. By reducing the fake-score updates per iteration from 5 (in the DMD2 baseline) to 1, SGMD achieves an approximate ~3x training speedup on Wan2.1-T2V-14B. In 4-step video generation, SGMD significantly improves motion intensity (OptFlow) and dynamic degree (DynDeg) compared to DMD2. This effectively mitigates the conservative updating and sluggish dynamics often caused by reverse-KL divergence.

## Weakness

1. While SGMD enhances video dynamics, its VBench Quality and Semantic scores are lower than those of DMD2 (e.g., Semantic score 75.64 vs. 77.46). Although the authors attribute this to the inherent differences between Fisher divergence and Reverse-KL, the lack of Human Evaluation makes it difficult to assess if the gain in dynamics fully compensates for the loss in visual and semantic fidelity. A real human evaluation is required to prove the claim of the authors.
2. The empirical comparison is strictly limited to DMD and SIM variants (DMD2, TSG-Fisher, TSG-SIM). Including broader comparisons with recent adversarial distillation or other non-score-matching video distillation methods would strengthen the evaluation.
3. This paper ignores a highly relevant paper: Flash-DMD[1], which also aims to increase the update frequency of fake scorer with a timestep-aware strategy, and accelerate the training speed to up to 20x of DMD2.
4. Accurate per-step training cost is required to measure the real training acceleration.

[1] Flash-DMD: Towards High-Fidelity Few-Step Image Generation with Efficient Distillation and Joint Reinforcement Learning

---

> ### Author Rebuttal · Authors · 2026-03-31
>
> We thank the reviewer for the important suggestions.
>
> ## W1
> We provide additional qualitative comparisons between DMD2 and SGMD in the supplementary material. These examples suggest that SGMD improves motion quality, while the trade-off on static visual quality is relatively modest. This is also consistent with the automatic metrics: the gap on VBench Quality is small, whereas SGMD brings substantially larger gains on FVD and dynamic-related metrics.
>
> Following the reviewer’s suggestion, we further conducted a human evaluation with pairwise comparisons between DMD2 and SGMD along four aspects: overall preference, motion quality, text-video alignment, and visual quality. The study involved 5 raters and 20 prompts, resulting in 100 pairwise judgments per aspect. The results are summarized below.
>
> | Evaluation Aspect | SGMD Wins (%) | Tie (%) | DMD2 Wins (%) |
> | --- | ---: | ---: | ---: |
> | Overall Preference | 65 | 22 | 13 |
> | Motion Quality | 71 | 5 | 24 |
> | Text-Video Alignment | 4 | 90 | 6 |
> | Visual Quality | 14 | 74 | 12 |
>
> We additionally evaluate with [VideoAlign](https://arxiv.org/abs/2501.13918), a human-feedback-trained reward model that provides a scalable proxy for human preference . While it does not replace human evaluation, it provides complementary evidence:
>
> | Metric | DMD2 | SGMD |
> | --- | ---: | ---: |
> | Overall | 18.86 | 19.36 |
> | Visual Quality | 8.47 | 8.19 |
> | Motion Quality | 4.15 | 4.99 |
> | Text Align | 6.24 | 6.19 |
>
> Together, the human evaluation and VideoAlign results suggest that SGMD improves motion quality while the degradation in text-video alignment and static visual quality is limited.
>
> ## W2
> We agree that broader empirical comparisons would strengthen the paper.
>
> Our current evaluation focuses on the most directly relevant baselines, namely DMD2, TSG-Fisher, and TSG-SIM, since SGMD is proposed to address a specific bottleneck in DMD-style bilevel distillation: the tracking lag between the fake scorer and the evolving generator. Accordingly, our goal is to isolate the effect of the proposed objective under the same teacher, the same 4-step setting, and the same prompt-only training regime.
>
> We also agree that adversarial distillation, consistency distillation, and other non-score-matching video distillation methods are important references. However, these methods typically involve different supervision, objectives, or architectural components, making it harder to attribute performance differences specifically to the tracking issue studied here. For this reason, we compare against the most directly comparable methods within the same distillation family.
>
> We will clarify this scope more explicitly in the revised paper, expand the discussion of broader video distillation methods, and, if feasible, include additional representative baselines.
>
> ## W3
> We thank the reviewer for pointing out Flash-DMD and agree that it is a highly relevant recent work. We will add the missing citation and discussion in the revised paper. Flash-DMD studies few-step image generation and improves the efficiency of DMD-style training through timestep-aware decoupling between diffusion matching and adversarial objectives, together with score-estimator stabilization and a joint RL refinement stage.
>
> Our work is closely related in motivation but different in focus. While Flash-DMD mainly improves training efficiency and perceptual fidelity through a timestep-aware training strategy, SGMD addresses the tracking-lag problem between the fake scorer and the evolving generator from the optimization perspective in few-step video distillation. In particular, SGMD reformulates the DMD-style objective via score gradient matching, a stable teacher stop-gradient Fisher objective, and decoupled updates, with the goal of improving training stability and temporal dynamics in video generation.
>
> Therefore, we view Flash-DMD as an important complementary related work rather than a redundant one. We will clarify this relationship more explicitly in the revised paper. Since Flash-DMD is developed in the image setting, a fully fair empirical comparison in the video setting would require nontrivial adaptation; nevertheless, we agree that such a comparison would be valuable and will include it if feasible.
>
> ## W4
> We respectfully disagree that the accurate per-step training cost is missing. This information is already reported in Appendix C, where we provide the per-step wall-clock training cost under the same experimental setting.
>
> More importantly, real training acceleration should be assessed from two complementary perspectives: (1) per-step computational cost, and (2) the total cost required to reach a competitive performance level. Appendix C explicitly reports the former, while our main experiments demonstrate the latter through improved convergence efficiency under the same training regime.
>
> We will make this point more explicit in the revised paper and reference Appendix C more prominently in the main text.

---

> > ### Author Rebuttal · Reviewer_XhXK · 2026-04-05
> >
> > The rebuttal has addressed most of my concerns, and I have raised my score accordingly. However, the Human Evaluation remains limited as it only covers 100 prompts. I recommend acceptance on the condition that the authors scale up the human study to at least 500 prompts and provide more methodological details in the final version. Additionally, all discussions and clarifications from the rebuttal must be incorporated into the camera-ready version.

---

### Decision · Program_Chairs · 2026-04-30

**Decision:**

Accept (regular)

**Comment:**

SGMD addresses a concrete bottleneck in DMD-style video distillation: tracking lag between the fake score network and the evolving generator. The dual-potential design (NR for outer-loop correction, RC for inner-loop contraction) reduces fake-score updates from 5 to 1 per iteration, yielding a ~3x training speedup on the 14B-parameter Wan2.1 model. Four-step distilled videos show substantially improved motion dynamics (OptFlow, DynDeg, FVD) over DMD2.

The main concern across reviewers was whether the motion gains compensate for modestly lower VBench quality/semantic scores. The authors addressed this with a human evaluation (65% overall preference for SGMD, 74% ties on visual quality) and VideoAlign scores, providing direct evidence that the quality trade-off is perceptually minor. Three of four reviewers found their concerns mostly or fully resolved (XhXK raised to 5; tJMk fully resolved at 4; DFbP "inclined to accept" at 4).

Reviewer yehJ's remaining concerns that the causal link between the algorithmic changes and motion improvement is indirect, and that the motivation around fake-score update frequency could be more explicit are reasonable requests for revision but do not identify fundamental flaws. The ablation evidence (TSG-Fisher already showing strong dynamics, SGMD preserving this while improving quality/semantic scores over TSG-Fisher) provides a reasonable, if indirect, chain of evidence.

The paper's scope is intentionally focused on DMD-style distillation rather than comparing across distillation families, which is a limitation but also a defensible choice given the goal of isolating the tracking-lag problem. The authors should incorporate the human evaluation, Flash-DMD discussion, and clarified motivation into the camera-ready version, and scale up the human study as Reviewer XhXK recommends.